# Incidence and Survival of Testicular Cancers in a Province in Northern Italy and Their Association with Second Tumors

**DOI:** 10.3390/biology12111409

**Published:** 2023-11-09

**Authors:** Lucia Mangone, Francesco Marinelli, Isabella Bisceglia, Cristina Masini, Andrea Palicelli, Fortunato Morabito, Stefania Di Girolamo, Antonino Neri, Carmine Pinto

**Affiliations:** 1Epidemiology Unit, Azienda USL-IRCCS di Reggio Emilia, 42122 Reggio Emilia, Italy; francesco.marinelli@ausl.re.it (F.M.); isabella.bisceglia@ausl.re.it (I.B.); 2Medical Oncology Unit, Azienda USL-IRCCS di Reggio Emilia, 42122 Reggio Emilia, Italy; cristina.masini@ausl.re.it (C.M.); stefania.digirolamo@ausl.re.it (S.D.G.); carmine.pinto@ausl.re.it (C.P.); 3Pathology Unit, Azienda USL-IRCCS di Reggio Emilia, 42122 Reggio Emilia, Italy; andrea.palicelli@ausl.re.it; 4Biotechnology Research Unit, AO di Cosenza, 87100 Aprigliano, Italy; f.morabito53@gmail.com; 5Scientific Directorate, Azienda USL-IRCCS di Reggio Emilia, 42122 Reggio Emilia, Italy; antonino.neri@ausl.re.it

**Keywords:** testis cancer, incidence, mortality, survival, histotype, second neoplasms

## Abstract

**Simple Summary:**

Testicular cancer is characterized by an excellent prognosis; long survival, however, requires careful monitoring, also due to the possibility of developing second tumors. Seminoma has a better prognosis, even many years after diagnosis, but also a greater possibility of developing second tumors. Non-seminoma presents an increased risk, especially among patients over 40.

**Abstract:**

This study investigated the incidence, mortality, and 5-year survival rates of testicular cancers diagnosed in a northern Italian province, which were eventually associated with previous or subsequent extratesticular neoplasms. Cases from 1996 to 2020 were examined by age and histotype (seminoma vs. non-seminoma). The standardized incidence rate was calculated using the European population, and the annual percent change (APC) was reported. The five-year relative survival was estimated using the Pohar Perme method. The association with the second neoplasm was also evaluated. In our study, 385 patients with testicular cancer were included, most of whom were aged between 30 and 40 years. The non-seminoma and seminoma groups accounted for 44% and 18% of younger adults, respectively. The incidence rate increased during the study period (APC 1.6*); however, it increased in seminomas (APC 2.3*) but not in non-seminomas (APC −0.1). Conversely, the mortality rate remained constantly low either overall or in each of the two groups. The overall 5-year survival rate of testicular cancer patients was 95% (99% and 88% for seminomas and non-seminomas, respectively). Primary extratesticular tumors were documented in 37 cases, 18 after and 19 before the testicular cancer diagnosis. Our study confirms that the increased incidence and excellent survival rate are the prerogative of seminomas.

## 1. Introduction

Primary testicular neoplasms are classically divided into germ cell tumors (GCTs) (seminomas and non-seminomas) and the less frequent non-GCT types (derived from the specialized stroma or sex cord cells); mixed tumors may also occur [1]. A cure is achievable for 95% of all patients with testicular tumors and 80% of those who have metastatic disease. Despite remarkable results with frontline and salvage combination chemotherapy, metastatic testicular cancer remains incurable in approximately 10% of patients, and novel treatment approaches and care strategies are warranted. In this context, it is necessary to evaluate the temporal trends of the epidemiological and clinical trends of seminoma and non-seminoma testicular cancers and the survivorship issues following treatment of these tumors [2,3]. For GCTs, the central European areas and northern areas constitute the territories with the highest incidence in the world (8–13 new cases × 100,000 inhabitants), together with the Maori populations in New Zealand, which constitutes the only non-European population with high incidence, while a low incidence is found in Africa, Asia, and the Caribbean (2 × 100,000) [4]. For several decades, the incidence rate of testicular cancer has been increasing in the US and many other countries, especially for seminomas, despite of the experts’ inability to explain the reason [5]. Testicular cancer is uncommon: about 1 in every 250 males will develop it at some point during his lifetime. This is largely a disease of young and middle-aged men (the average age is around 33 years), with about 6% of cases occurring in children and teens and about 8% in men older than 55 years [5]. Undescended testicles (cryptorchidism), family history of testicular cancer, age, environmental exposures, and racial features are all risk factors implicated in the genesis of testicular cancer [6,7,8]. Physical activity has also been implicated as a risk factor, but the relationship remains controversial [9]. A low socioeconomic status of the mother also seems to play a role in the genesis of these neoplasms, although studies are limited [10,11]. Testicular cancer survival in the general population is more likely to be attributable to healthcare system factors than to the personal or socioeconomic factors of the men themselves [12]. In Italy, testicular cancer accounts for roughly 2300 new cases per year, representing approximately 1% of all male malignancies [13]. It is the most prevalent neoplasm in males under 50 years and accounts for 12% of all incident cases, whereas it is a very uncommon neoplasm among men over 50. The incidence trend appears to be significantly increasing (+2% per year) and shows slightly higher rates in Northern Italy than in Central and Southern Italy (7.6, 7.1, and 6.4 × 100,000, respectively). The 5-year relative survival of testicular cancer in Italy is 93%, with no geographical differences [14].

This study aimed to examine the incidence and mortality trends and the 5-year relative survival rate of testicular cancer by morphology in a northern Italian province and investigate their association with second tumors.

## 2. Materials and Methods

Testicular cancer cases diagnosed in 1996–2020 were defined as topography C62 [15] based on the *International Classification of Diseases for Oncology*, Third Edition (ICD-O-3). The population covered by the Reggio Emilia Cancer Registry (RE-CR) amounts to 532,000 inhabitants. The leading information sources of the RE-CR are anatomic pathology reports, hospital discharge records, and mortality data, integrated with laboratory tests, diagnostic reports, and information from general practitioners. In particular, the RE-CR algorithm identifies suspected cases by combining these three traditional databases (pathological reports, hospital discharge archives, and mortality records) with general and biomolecular laboratory tests according to a deterministic list of diagnoses. All suspected cases are accompanied by a morphological code that comes from a pathological laboratory (SNOMED and ICD-O-3) or, when not present, from discharge records (ICD-9-CM) or the cause of death (ICD-10). The strengths of RE-CR are a high percentage of microscopic confirmation (94% for testicular cancer) and a low rate of DCO (Death Certificate Only) cases that is below 0.1% [16]. It is also one of the few Italian CRs to have already published data updated to 2020. The Cancer Registry collects data and information following current flows to produce incidence, mortality, prevalence, and survival statistics for the resident population and demographic subgroups as required by the epidemiological report, which is defined by Law no. 29 of 22 March 2019 that regulates the cancer registries in Italy. The law exempts the registries from collecting informed consent. The procedures for conducting epidemiological analyses of the Reggio Emilia Cancer Registry data were approved by the provincial Ethics Committee of Reggio Emilia (Protocol no. 2014/0019740 on 8 April 2014). The provincial Ethics Committee of Reggio Emilia approved the study on 4 August 2014 (Protocol no. 2014/0019740).

Descriptive analyses of the characteristics of patients with germ cell tumors were presented by histotype (seminoma and non-seminoma). Since the average age of testicular cancer is 33 years [5], we divided the age groups into 3 groups: <30, 30–40, and 40+, and years of study were divided into 5 periods (1996–2000; 2001–2005; 2006–2010; 2011–2015; 2016–2020). For the two histotypes of testicular cancer, age-standardized incidence and mortality rates (SR) were calculated using the Province of Reggio Emilia population (recorded on 1 January of each year) as denominators. We considered the last 25 years (1996–2020). The direct method of standardization was applied to adjust rates for age using the 2013 European Standard Population as a reference. Trends over time were analyzed by calculating the annual percent change (APC) in age-standardized rates using Joinpoint Regression for seminoma and non-seminoma groups. The 5-year relative survival was calculated for tumors incident in the period of 1996–2015 and was divided by seminoma and non-seminoma and by age groups. The five-year relative survival rate was estimated using the Pohar Perme method; with this estimator, net survival for a cohort is estimated by weighting by the inverse of the individual-specific expected survival probabilities. The weights inflate the observed person-time and number of deaths to account for person-time and deaths not observed because of mortality due to competing causes [17]. Relative survival, calculated as an estimate of net survival, representing cancer survival, in the absence of other causes of death, is defined as the ratio of the proportion of observed survivors in a cohort of cancer patients to the proportion of expected survivors in a comparable set of cancer-free individuals. The age-specific incidence rate for five-year classes was calculated for seminoma and non-seminoma. Independent tumors were calculated by examining medical records and evaluating the presence of a second malignant neoplasm that occurred in the period of study, either before or after diagnosis of testicular cancer. We also reported the median time (in years) between the onset of testicular cancer and the second tumor inception. We also calculated the odds ratio (OR) with a relative 95% confidence interval (CI) using logistic regression analysis to assess the impact of other cancers on possible predictors, i.e., histotype and age at diagnosis. Furthermore, we performed a Cox regression model to investigate the association between histotype, age, and overall survival (time was expressed in years). The time-to-event analysis (overall survival) by histotype was performed using the Kaplan–Meier method. Analyses were performed using STATA 16.1 software. In this study, we reported 95% CIs. These analyses can be replicated with other cancer registries.

## 3. Results

Three hundred eighty-five patients with testis cancers were identified, of whom 63.4% had seminomas and 36.6% non-seminomas (Table 1); the average age of seminomas was 39.7 years (SD = 11.8) while for non-seminomas, it was 33.8 years (SD = 14.2).

Most seminoma cases (about 42%) were diagnosed between 30 and 40 years, only 18% were diagnosed before 30 years, and roughly 40% were diagnosed beyond 40 years (Table 1). Conversely, among non-seminomas, only 20% of cases were detected beyond the age of 40 years, about 36% were between the ages of 30 and 40 years, and 44% of cases were diagnosed before the age of 30 years (Table 1). The age-specific incidence rate showed the trend of seminomas and non-seminomas by age (Figure 1): the former reached a peak around the age of 30 years and then dropped sharply at later ages.

The latter group, however, presented a bimodal trend with a first peak around the age of 20 years and a second peak around the age of 30 years.

Regarding the diagnosis period, the number of cases seemed to be increasing in the last two five-year periods (24.4% in 2011–2015 and 22.6% in 2016–2020) compared to the first one (15.6% in 1996–2000). This increase, however, was almost entirely due to seminomas, which went from 14.8% in the first five-year period to 24.6% in the last period (Table 1). The trends in incidence and mortality over the considered 25 years are shown in Figure 2.

The yearly incidence of seminomas increased by 2.3% according to the 25-year recording trends, as the Standardized Rate (SR) was 1.5 × 100,000 in 1996 and 4.2 × 100,000 in 2020; conversely, there was no discernible change in the incidence of non-seminomas (SR 2.5 × 100,000 in 1996 and 2.2 × 100,000 in 2020). The rate of mortality remained quite low throughout the whole timeframe under consideration.

The 5-year survival rate of our testicular cancer patients confirmed the good prognosis of this neoplasm: overall survival accounted for 95% (95% CI 91–97) but with a difference between seminomas (99%, 95% CI 89–100) and non-seminomas (88%; 95% CI 80–93) (Table 2).

The survival rate was 97% (95% CI 91–99) for the 30–40 age group, was reduced to 95% (95% CI 87–98) for men under 30 years of age, and decreased to 91% (95% CI 78–96) for the age group over 40 years. The good prognosis of seminomas, compared to non-seminomas, remained unchanged even many years after diagnosis (Figure 3).

Considering the two histotype groups, survival has been confirmed to be very high for seminomas, with values reaching 100% for patients under 30 years of age, and accounting for 99% and 98% in the 30–40 and >40 year age groups, respectively. Among non-seminomas, however, survival remained high for the <30 and 30–40 year age groups (91% and 93%, respectively) but it dropped to 64% for men over 40 years of age.

Since testicular cancer has high survival rates, it is also important to evaluate the arising of further independent tumors during the follow-up of these young patients. Table 3 reports that among the 385 patients diagnosed with testicular cancer, 19 (5%) had been previously diagnosed with an infiltrating malignant tumor, mainly arising from the skin (*n* = 5; non-melanomas) or representing hematological tumors (*n* = 3), followed by bladder, colon, liver, and lung cancers (*n* = 2 cases for each primary site).

In only one case, there was a previous diagnosis of nasal cavity cancer; he was a patient who had already had a multiple diagnosis of bladder cancer and hematological tumors. Finally, a patient over 80 years of age revealed a personal oncologic history of colon, prostate, and skin cancer before the arising of the testicular primary.

On the other hand, after the diagnosis of testicular cancer, 18 patients (5%) developed a second neoplasm, mainly involving the skin (4 cases), lung (4 cases), and prostate (3 cases), with times since the first diagnosis of 9, 8, and 7 years, respectively. No contralateral testicular tumors were diagnosed.

Since histotype and age are the two most important determinants in the prognosis of the disease, we added a multivariate analysis. After adjusting for age at diagnosis, logistic regression analysis revealed that among seminoma patients, the odds of developing a second cancer increased by about twice [OR = 1.9 (95% CI 0.7–5.4)]. Notably, the odds ratio peak was reached in patients of over 40 years [OR = 8.4 (95% CI 1.9–37.8)] (Table 4).

## 4. Discussion

Our study aimed to examine the incidence and mortality trends, as well as the 5-year relative survival, of testicular tumor diagnoses in a northern Italian province, and their associations with the previous or subsequent development of further neoplasms. The literature data show that testicular cancer is rare. In the USA, it represents 1% of tumors, with an incidence rate that has increased from 5.7 cases per 100,000 in 1992 to 6.8 cases per 100,000 in 2009 [18,19]. A European population study, which analyzed data from the period of 1970–2012, reported an average annual testicular cancer rate of 7.32 per 100,000, with a non-significant increasing trend during the study period [20]; despite early diagnosis, 14% of patients presented at an advanced stage, typically being men with a diagnosis of non-seminoma and low social-economic status. According to our findings, 63.4% of the 385 diagnosed cases were seminomas and 36.6% were non-seminomas. The former type was more frequent in adults over 40 years of age, while the non-seminoma cases were more frequent in young men. The distribution over the years shows a slight increase among non-seminomas, which is confirmed by the incidence trends over the entire period. The age distribution confirms a younger peak for non-seminomas (15–20 years) than for seminomas (30–35 years); for both groups, the peak occurred at an earlier age than previously reported in the literature (with a peak at 25–29 years for non-seminomas and at 35–39 years for seminomas) [21]. The 5-year survival rate for testicular cancer was extremely high (95%) in the USA [22]; this high value was predominantly due to the seminomas’ rate (99%), but it dropped to 88% for non-seminomas. There were no significant differences in survival by age in our study, but when looking at seminomas and non-seminomas separately, the non-seminoma survival rate dropped to 64% in the men over 40, as reported in other studies [23]. Nevertheless, the availability of standardized treatments after the 1980s correlated with a significantly lower overall death rate [24,25,26]. Because of the increased incidence of long-term survivors due to the survival rate improving over the years (from 83% in 1975–1977 to 97% in 2005–2011) [18], the likelihood of developing a second neoplasm is a quite expected phenomenon [27,28]. According to a study conducted by the Swiss Cancer Registry, 9.4% of patients developed a second cancer with a relative risk (RR) of 1.30 (95% CI: 1.20–1.40), including a high risk for solid tumors, non-Hodgkin lymphoma, and leukemia [29]. Schaffar also reported a more than doubled risk of second tumors for the contralateral testis, bladder, and pancreas [20]. Travis et al. [30] highlighted that survivors of testicular cancer had a statistically significantly increased risk of solid tumors for at least 35 years after treatment. The incidence was 36% and 31% for seminoma and non-seminoma cases, respectively, compared to 23% among the general population. Young patients may be exposed to higher risk levels once they reach older ages. Curreri et al. reported that the risk may be even higher for subjects treated with high-dose radiotherapy [31]. In one study [32], after a median of 80 months of follow-up, 11% of patients with stage I seminomas had a recurrence, which occurred in the retroperitoneum in 86% of cases. In another study, 70% of non-seminoma stage I patients presented recurrences within 6 months that were localized in the abdomen (59% of cases), lung (16%), or inguinal regions (5%) [33].

The problem of recurrence is one of the most important issues to address in the management of patients with testicular cancer; in recent years, the COVID-19 pandemic has emphasized the fact that, although only active surveillance of stage I seminomas has the same overall survival as adjuvant therapy, treatment of these patients would have a greater protective effect in terms of recurrence-free survival [34].

The most frequent second tumor sites included the stomach, pancreas, pleura, bladder, colon, and esophagus [31] and also the skin (melanoma), thyroid, kidney, bladder, and connective tissue [35]. In our study, few extratesticular tumors were observed, also considering the young age of these patients and their shorter follow-ups. However, 19 patients had already had a neoplasia antecedent to the diagnosis of the testicular primary (in the skin, hematological tumors, bladder, colon, liver, and lung). The 18 tumors that appeared after the diagnosis of testicular cancer affected sites such as skin, lung and prostate. Although testicular cancer is a rare neoplasm, the availability of 25-year incidence rates allowed us to collect 385 cases for the study; in our opinion, this data quality is good, considering that 99% of cases were histologically confirmed. Another point of strength of our study is that it looked at population data rather than hospital data. The study’s weakness is the limited number of cases in specific subgroups, which prevented us from providing many details, as well as the lack of information on treatment schedules.

## 5. Conclusions

Overall, our data confirmed the steadily rising incidence of this still-rare cancer. Fortunately, substantial advances in treatment have occurred over the last few decades, making testicular cancer one of the most curable malignancies. However, because testicular cancer typically occurs in younger men, considerations of the impact of treatment on fertility, quality of life, and long-term toxicity are paramount; an individualized approach must be taken with patients based on their clinical and pathologic findings.

## Figures and Tables

**Figure 1 biology-12-01409-f001:**
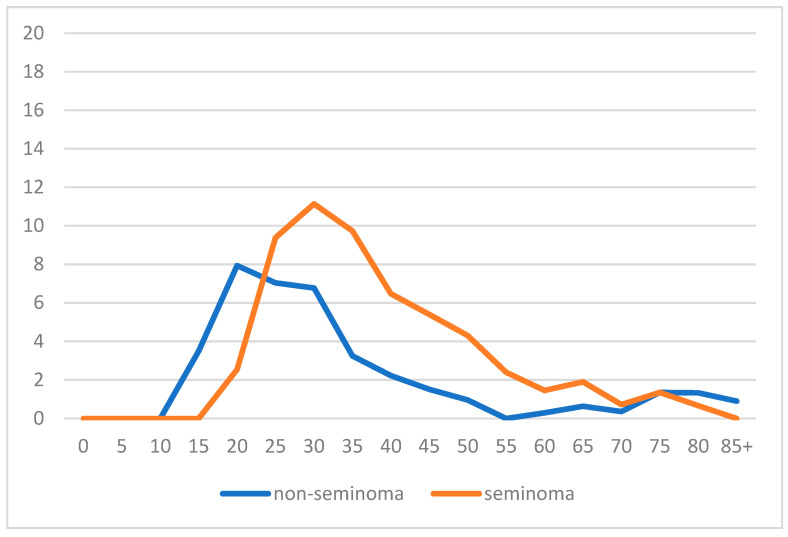
Reggio Emilia Cancer Registry. Years 1996–2020. Age-specific incidence rate by histotype.

**Figure 2 biology-12-01409-f002:**
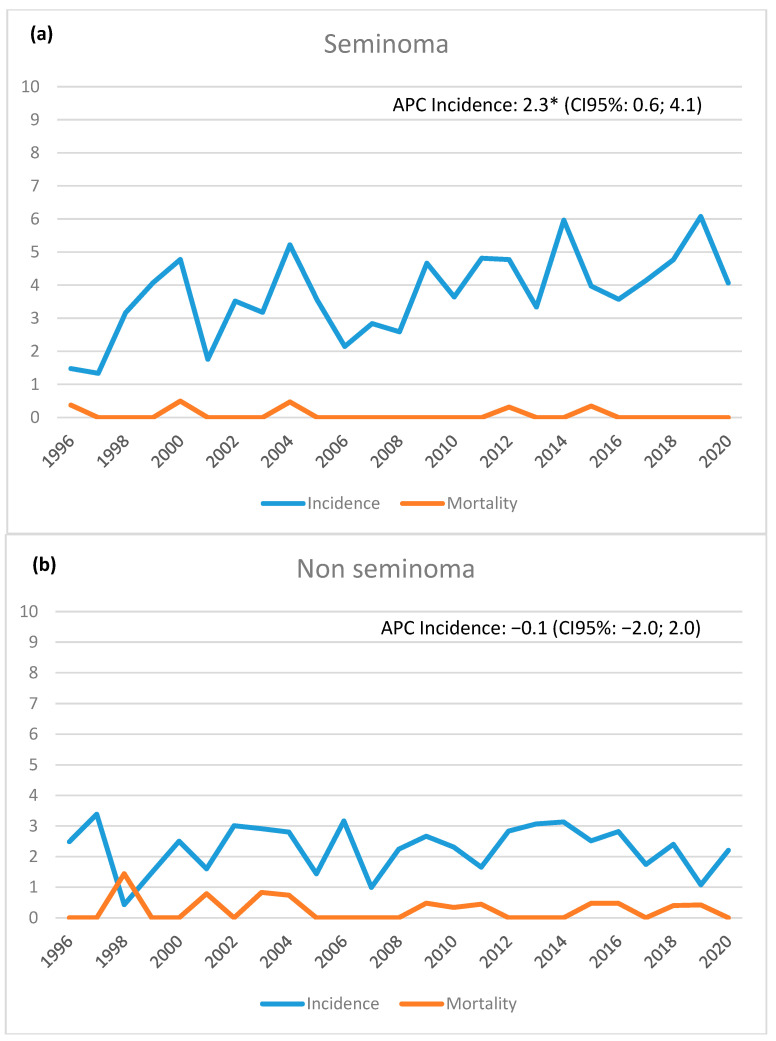
Reggio Emilia Cancer Registry. Years 1996–2020. Incidence and mortality rates per 100,000 p-y by histotype ((**a**) seminomas; (**b**) non-seminomas). * statistically significant.

**Figure 3 biology-12-01409-f003:**
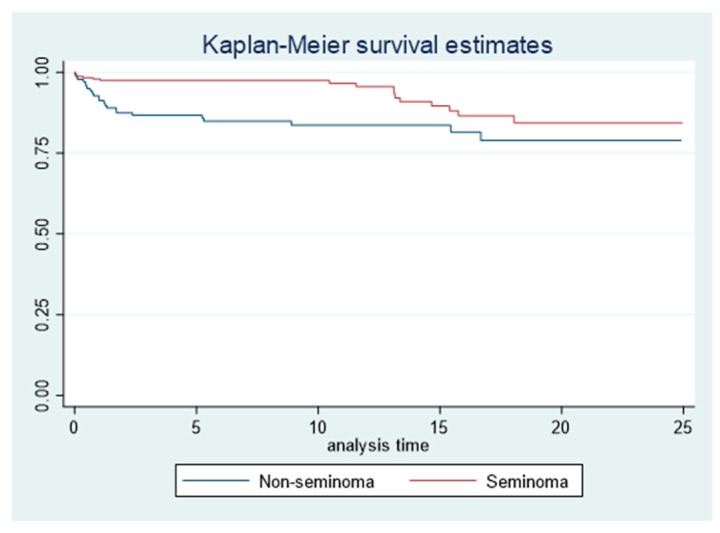
Overall survival probabilities by histotype. Cox proportional hazards model adjusted for histotype and age. CI: confidence interval; HR: hazard ratio.

**Table 1 biology-12-01409-t001:** Reggio Emilia Cancer Registry. Years 1996–2020. Number of cases by age and period.

	Non-Seminoma	Seminoma	Total
	*n* (%)	*n* (%)	*n* (%)
Overall	141 (36.6)	244 (63.4)	385 (100)
Age at diagnosis			
<30	62 (44.0)	44 (18.0)	106 (27.5)
30–40	51 (36.2)	102 (41.8)	153 (39.7)
40+	28 (19.9)	98 (40.2)	126 (32.7)
Period of diagnosis			
1996–2000	24 (17.0)	36 (14.8)	60 (15.6)
2001–2005	30 (21.3)	42 (17.2)	72 (18.7)
2006–2010	28 (19.9)	44 (18.0)	72 (18.7)
2011–2015	32 (22.7)	62 (25.4)	94 (24.4)
2016–2020	27 (19.1)	60 (24.6)	87 (22.6)

**Table 2 biology-12-01409-t002:** Reggio Emilia Cancer Registry. Years 1996–2015. Five-year relative survival by age and histotype.

	Non-Seminoma			Seminoma			Overall		
	*n*	%	95% CI	*n*	%	95% CI	*n*	%	95% CI
Overall	114	88	80–93	184	99	89–100	298	95	91–97
Age at diagnosis (years)									
<30	53	91	79–96	35	100	100–100	88	95	87–98
30–40	44	93	80–98	79	99	84–100	123	97	91–99
40+	17	64	32–84	70	98	70–99	87	91	78–96

**Table 3 biology-12-01409-t003:** Reggio Emilia Cancer Registry. Years 1996–2020. Independent tumors before and after a diagnosis of testicular cancer.

Before			After		
**Site**	** *n* **	Testicular cancer	**Site**	** *n* **	**Median Age (Years)**
Skin	5		Skin	4	9.4
Hematological tumors	3		Lung	4	8.5
Bladder	2		Prostate	3	6.7
Colon	2		Hematological tumors	1	15.8
Liver	2		Pancreas	1	13.8
Lung	2		Thymus	1	3.3
Nasal cavity	1		Connective tissue	1	3.6
Bladder and hematological tumors	1		Bladder	1	7.5
Skin, prostate, colon	1		Central nervous system	1	8.4
			Thyroid	1	0.4
Total	19		Total	18	7.5

**Table 4 biology-12-01409-t004:** Associations between second tumor and histotype and age. Odds ratio (OR) with 95% confidence interval (CI).

Characteristics	Univariate Analysis		Multivariate Model	
	OR	95% CI	OR	95% CI
Histotype				
Non-seminoma	1	Ref.	1	Ref.
Seminoma	3.0	1.1–8.0	1.9	0.7–5.4
Age at diagnosis				
<30	1	Ref.	1	Ref.
30–40	2.1	0.4–10.7	1.8	0.4–9.3
40+	10.4	2.4–45.5	8.4	1.9–37.8

## Data Availability

The data presented in this study are available on request from the corresponding author. The data are not publicly available due to ethical and privacy issues; requests for data must be approved by the Ethics Committee after the presentation of a study protocol.

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
