# Peer review of "Incidence and Survival of Testicular Cancers in a Province in Northern Italy and Their Association with Second Tumors"

_biology, 2023, doi:10.3390/biology12111409_

Round 1

Reviewer 1 Report

Comments and Suggestions for Authors

The authors herein investigate "the incidence, mortality, and 5-year survival of testis cancer by morphology (Seminoma versus non-Seminoma) in a northern Italian province" and evaluate the incidence and type of separate neoplasms occurring in a subset of patients.

The study is fairly well-written and easy to read.

I do not identify any major issue, but have a few minor observations and corrections, as follows:

Line 92: The authors state that "we have divided the age groups into 4 groups: <30; 30-40, and 40+[...]". The groups should therefore be three, not four.  I would also refrain from changing the verb tense within the manuscript ("we divided" or better "patients were stratified in", rather than "we have divided").

In general, numbers in the text should be spelled out if lesser or equal ten: e.g., "5 groups" becomes "five groups" etc.

Line 30-31 (and elsewhere in the text): The terminology used to describe testicular neoplasms could be improved by using the word "type" or "subtype" rather than "forms", and the term "germ cell tumors" or "germ cell-derived tumors" rather than "germinal forms". Including a pathologist in the authors' roster (or running the manuscript by one of them for critical review), or simply looking at [modern] pathology literature and classification of diseases, might be beneficial to the authors, for polishing the nomenclature.

Along the same line, despite the introduction correctly mentioning several "types" of testicular tumors, the study appears to evaluate statistics of germ cell tumors only (seminomatous versus non-seminomatous).  It should be clarified at some point in the materials and methods that the tabulated results do not include sex cord stromal tumors or any other tumor of the testis except germ cell tumors (seminomatous and non-seminomatous).

Finally, the term "second neoplasia" is slightly misleading: we often use the term "secondary" neoplasia to indicate a tumor that arises "after" the first one. The authors seem to use the term interchangeably for "independent"/"separate" neoplasms that occurred in a subset of the study population either before OR after the testicular tumor. I would suggest changing the qualifier to either "independent" or "separate", in lieu of "second"

Comments on the Quality of English Language

Already commented upon (see "Comments and Suggestions for Authors
")

Author Response

Thanks to the reviewer for his valuable comments. We hope that our fixes are satisfactory.

Best regard, 

Lucia Mangone

Reviewer 2 Report

Comments and Suggestions for Authors

I enjoyed reading the authors' findings, which are very interesting and valuable. Testis cancer is not a high incidence disease, but it is a significant cancer in the realm of urology.

1. in Table 2, regarding 5-year survival, you were not able to obtain a Kaphlan Meier Curve or Cox harzard regression? If you were unable to do so due to the rarity of the cancer and the difficulty in setting up the data, I think it would be necessary to explain the limitation. 

2. Due to COVID-19, there was a change in the EAU recommendation from 2020-2022. Please mention this and cite the paper below and explain further.

http://dx.doi.org/10.3390/medicina58111514

Author Response

Thanks to the reviewer for his valuable comments. We hope our fixes are satisfactory.

Best regards,
Lucia Mangone
